# Carba-NAD binding activates SIR2 by reshaping conformational plasticity and rewiring long-range allosteric networks

**Bao-Dan Zhang**[1,2,3], **De-Rui Zhao**[1,2,3], **Meng-Ting Liu**[1,2], **Li-Quan Yang**[1,2,3]*, **Peng Sang** [1,2,3]*

**1** College of Agriculture and Biological Science, Dali University, Dali, China, **2** Key Laboratory of Bioinformatics and Computational Biology of the Department of Education of Yunnan Province, Dali University, Dali, China, **3** Co-Innovation Center for Cangshan Mountain and Erhai Lake Integrated Protection and Green Developmentof Yunnan Province, Dali University, Dali, China

* ylqbioinfo@dali.edu.cn (L-QY); pengsang@dali.edu.cn (PS)

## Abstract

Allosteric regulation enables proteins to couple local structural changes to distal functional outcomes, yet the underlying mechanisms often remain difficult to fully decipher. Using yeast SIR2, an $NAD^+$-dependent deacetylase, as a model system, this study systematically elucidates how cofactor binding reshapes its conformational dynamics and internal communication network. Through multiple 3-μs molecular dynamics simulations combined with a graph-based deep learning model (Neural Relational Inference), we identify a highly reproducible characteristic response across independent replicates: the β1–α2 loop near the active site undergoes pronounced rigidification, whereas several distal structural modules exhibit concomitant increases in flexibility, together forming a "core-locking with peripheral-release" dynamic mode. Further signal-pathway analysis reveals that the local and distal conformational changes are not independent; instead, they are interconnected through newly identified "relay-type" residues such as Pro214 and Thr224. These residues act as bridges, converting the previously β1–α2-centered centralized network into a relay-style network coordinated by multiple nodes, thereby establishing a continuous and directionally coherent allosteric cascade. Beyond mechanistic insights, we also identify a distal cavity spatially overlapping with key relay residues, whose physicochemical properties meet the criteria of druggable pockets. This structural convergence suggests that future small-molecule allosteric activators may exploit this intrinsic communication pathway to mimic or amplify the regulatory effects of the cofactor $NAD^+$. Given that $NAD^+$ levels decline with aging, this cavity provides a rational target for designing longevity-promoting allosteric activators.

**Data availability statement:** The experimental data, MD simulation trajectories and parameter files, as well as the custom scripts used to perform the analyses in this study have been made publicly available in the Zenodo database at: https://doi.org/10.5281/zenodo.17247434.

**Funding:** This study was supported by the Xingdian Talents Support Program of Yunnan Province (YNWR-QNBJ-2020-086 to PS) and the National Natural Science Foundation of China (31860243 and 31960198 to PS). The funders had no role in study design, data collection and analysis, decision to publish, or preparation of the manuscript. No authors received a salary from the funders.

**Competing interests:** The authors declare no conflicts of interest in this work.

## Author summary

Sirtuins (SIR2) are a family of longevity-associated proteins that play key roles in aging, metabolism, and genome stability, with their enzymatic activity dependent on the cofactor $NAD^+$. Although their static structures have been resolved, how $NAD^+$ triggers activation in distal regions remains unclear. In this study, we combine long-timescale molecular dynamics simulations with Neural Relational Inference (NRI) to elucidate the intramolecular signal transduction mechanism following the binding of the $NAD^+$ analog Carba-NAD. We find that cofactor binding stabilizes the catalytic region while increasing the flexibility of distal regions, thereby reshaping the energy landscape and promoting the formation of activation-prone conformations. More importantly, the network of residue–residue interactions is reorganized, with key residues acting as bridges that facilitate long-range signal propagation. These findings provide new insights into the molecular regulatory mechanism of SIR2 and offer potential strategies for drug development targeting aging-related diseases.

## 1. Introduction

Sirtuins (the SIR2 protein family) are a class of $NAD^+$-dependent deacetylases that play essential roles in chromatin silencing, genome stability, metabolic regulation, and aging [1–3]. Initially identified in yeast as transcriptional repressors, SIR2 proteins were later shown to sense intracellular $NAD^+$ levels and couple metabolic state to gene expression through lysine deacetylation [4]. The structure of yeast SIR2 closely resembles that of its mammalian homologs and adopts a characteristic bilobal architecture: a Rossmann-fold domain responsible for recognizing and positioning $NAD^+$, and a $Zn^{2+}$-binding domain that provides structural support via a conserved $Cys_4$ coordination motif. Together, these domains form a deeply recessed catalytic cleft [5–7]. Despite extensive crystallographic and biochemical studies, the precise molecular mechanism by which $NAD^+$ binding promotes catalytic activation of SIR2 remains incompletely understood.

Structurally, the two core domains are connected by four conserved loop regions that form a deep catalytic channel, into which the acetyl-lysine substrate and $NAD^+$ approach the central reaction site from opposing sides (Fig 1a). The Rossmann domain [8] positions $NAD^+$ such that its nicotinamide moiety faces the catalytic residues, while the $Zn^{2+}$-binding domain [9] provides mechanical stability. Several flexible loop elements surround the channel and play crucial roles in regulating conformational transitions. Among them, the β1–α2 loop (residues 35–63), located adjacent to the cofactor-binding pocket and exhibiting notable conformational variability, has been widely implicated in substrate recognition, catalytic regulation, and product release [10].

Because native $NAD^+$ is rapidly consumed during the reaction and difficult to capture in stable crystal complexes, many structural studies employ the non-hydrolyzable

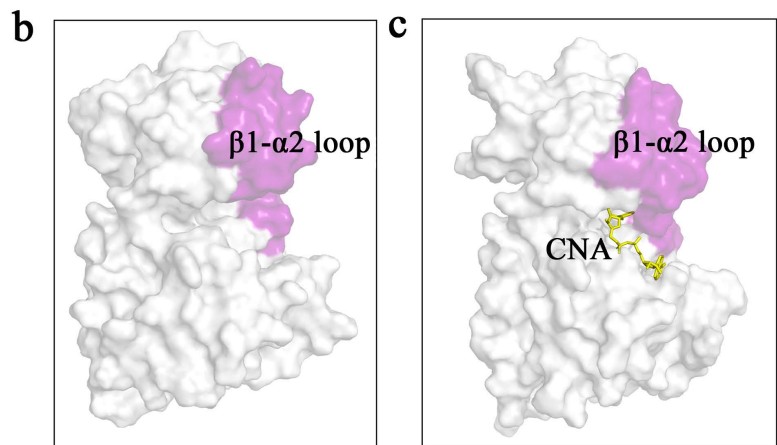

**Fig 1. Structural features of SIR2 in the apo and CNA-bound states.** (a) Cartoon representation of the SIR2 monomer and its complex with CNA (PDB ID: 1SZC). The Rossmann-fold domain is shown in cyan, and the β1–α2 loop is highlighted in purple, with the annotated cofactor-binding loop corresponding to a segment within this region. Connecting loops are shown in orange. The acetyl-lysine peptide substrate (green), the CNA molecule (yellow), and the zinc ion (red sphere) are also depicted. (b, c) All structural renderings are shown in the same molecular orientation as in panel (a). Surface views of the catalytic cleft of SIR2 in the apo state (b) and CNA-bound state (c), highlighting the β1–α2 loop (purple) and CNA (yellow). Cofactor binding induces an inward shift and rigidification of this loop, thereby narrowing the cleft and stabilizing the catalytically active conformation.

NAD⁺ analog Carba-NAD (CNA). CNA stably occupies the natural NAD⁺ binding site, making it an ideal surrogate for elucidating cofactor-binding modes. Existing SIR2 crystal structures show that between the apo, substrate-bound, and CNA-bound states, the β1–α2 loop switches among distinct conformations accompanied by local rearrangements of the catalytic cleft (Fig 1b and 1c). Although the catalytic core of sirtuins is highly conserved, their N- and C-terminal regions vary substantially in length and sequence, giving rise to subtype-specific substrate preferences and subcellular localizations [11–13]. Given that intracellular NAD⁺ concentrations fluctuate with metabolic state and aging, understanding how NAD⁺ reshapes the conformational ensemble of SIR2 is critical for elucidating its functions in redox regulation, chromatin dynamics, and aging.

However, comparisons of static structures remain insufficient to explain the full landscape of SIR2 activation. Although previous studies have observed subtle changes around the β1–α2 loop upon cofactor binding, a central question remains

unresolved: how can such limited local perturbations drive large-scale functional activation of the entire SIR2 protein? Unlike classical allosteric systems that rely on pronounced rigid-body motions [14], sirtuins tend to achieve allosteric effects through partial ordering, flexibility modulation, and internal information transfer [15]. Yet the logic underlying this "local micro-perturbation → distal regulatory response" relationship has not been systematically clarified. Cofactor binding induces only minor structural shifts around the catalytic cleft, yet it consistently enhances enzymatic activity and alters the dynamic properties of multiple distal regions, implying the presence of hidden communication pathways within SIR2. The challenge lies in understanding how a local anchoring event can trigger a distributed global response in the absence of obvious domain-scale motions.

To account for such long-range effects, several models have been proposed—including conformational selection, induced fit, and entropy-driven regulation [16,17]—but none fully explain the spatial correlation and coordination among distinct regions. More importantly, the lack of a dynamic, system-level perspective has long hindered efforts to elucidate how SIR2 converts a single cofactor-binding event into a global structural reorganization. Understanding this process requires tools capable of capturing dynamic residue–residue interactions and tracing how signals propagate across the protein structure—features that static crystal structures fundamentally cannot provide.

To address this gap, we integrate long-timescale molecular dynamics (MD) simulations [18,19] with a graph-neural-network–based Neural Relational Inference (NRI) model [20,21] to systematically characterize the dynamic response of SIR2 following CNA binding [10,22]. This combined approach enables simultaneous monitoring of changes in conformational flexibility and reconstruction of the time-evolving residue communication network. By tracking how the rigidification of the β1–α2 loop—an initially local perturbation—propagates along the structural framework toward distal regions, we aim to reveal how cofactor recognition redistributes internal dynamics, rewires allosteric communication pathways, and promotes the formation of activation-prone conformational states.

Collectively, this work provides a system-level view of signal propagation within SIR2 and establishes a structural foundation for designing allosteric regulatory strategies targeting aging- and metabolism-related diseases.

## 2. Materials and methods

### 2.1. Structure preparation

The crystal structure of the yeast SIR2–Carba-NAD (CNA) complex was obtained from the Protein Data Bank (PDB ID: 1SZC) [10,23]. This structure contains the full coordinates of CNA but lacks residues Ser210–Pro214 in chain A. To generate a complete apo-state model, we performed homology modeling using SWISS-MODEL [24,25] based on the 1SZC sequence, and selected the top-scoring model according to GMQE and QMEAN metrics for subsequent simulations and analyses.

To construct the CNA-bound complex, we carried out molecular docking of CNA. CNA is a stabilized NAD⁺ analog in which the ribose oxygen atom is replaced by carbon, allowing it to retain the native binding mode. Docking was performed with AutoDock Vina [26]: polar hydrogens were added to the receptor model during preprocessing, and Gasteiger charges were assigned. The docking grid was centered at (–10.376 Å, 34.764 Å, –41.484 Å) with a box size of 40 Å and a grid spacing of 1 Å. The exhaustiveness level was set to 16 to ensure sufficient conformational sampling. Docked poses were ranked by binding energy, and the lowest-energy pose was selected to construct the CNA-bound model. CNA parameters were generated using the CHARMM General Force Field (CGenFF) for downstream MD simulations. Parameters and topology for $Zn^{2+}$ were automatically generated through the CHARMM-GUI [27,28] metals modeling workflow under the CHARMM36 force field, and integrated consistently with the protein topology during system construction.

### 2.2. MD simulations

We performed all-atom MD simulations for two systems: apo-SIR2 generated from the homology-completed structure, and the CNA–SIR2 complex obtained through AutoDock Vina docking. All simulations were carried out using GROMACS

2022.5 [29] with the CHARMM36 force field [30]. CNA parameters were generated via CGenFF [31] and incorporated into the topology, and $Zn^{2+}$ parameters were obtained from CHARMM-GUI and integrated consistently.

All systems were placed in a dodecahedral box and solvated with TIP3P water [32], with a minimum solute–box distance of 1.2 nm. $Na^+$ and $Cl^-$ ions were added to neutralize the system and adjust the ionic strength to 100 mM. Energy minimization was performed using the steepest descent algorithm [29] until the maximum force was below 1000 $kJ \cdot mol^{-1} \cdot nm^{-1}$. This was followed by 100 ps of NVT equilibration and 100 ps of NPT equilibration at 300 K and 1 bar, during which positional restraints of 1000 $kJ \cdot mol^{-1} \cdot nm^{-2}$ were applied to all backbone heavy atoms.

For production simulations, each system was simulated with three independent 3-μs trajectories, giving a total sampling time of 9 μs. To ensure statistical independence, each trajectory was initiated by reassigning velocities after the initial equilibration phase, followed by an additional set of 100 ps NVT and 100 ps NPT equilibration using the same temperature, pressure, and restraint conditions. Production runs of 3 μs were then started from these independently equilibrated states rather than simply altering the random seed for velocity generation.

A 2-fs timestep was used, and bond constraints were applied using the LINCS algorithm [33]. Long-range electrostatics were treated with the Particle Mesh Ewald (PME) method [34] (real-space cutoff 1.0 nm, Fourier spacing 0.16 nm). van der Waals interactions used a 1.0 nm cutoff. Temperature was maintained using the velocity-rescale thermostat ($\tau = 0.1$ ps) [35], and pressure was controlled with the Parrinello–Rahman barostat ($\tau = 2$ ps) [36]. Trajectories were saved every 10 ps for downstream analyses [37].

### 2.3. Structural and dynamic analysis

To systematically evaluate the conformational stability and dynamical properties of SIR2 in the apo and CNA-bound states, we performed multi-level structural analyses on the simulation trajectories using GROMACS. First, the backbone root-mean-square deviation (RMSD) was calculated with 'gmx rms', where each of the three independent trajectories was least-squares aligned to its own initial structure to assess global structural stability. The per-residue root-mean-square fluctuations (RMSF) of backbone Cα atoms were computed using 'gmx rmsf', adopting the same backbone alignment protocol as in the RMSD analysis.

We used the gmx mindist tool to quantitatively analyze the number of native contacts (NNCs). During the analysis, only heavy atoms of the entire protein were considered. A contact was defined as being formed when the minimum interatomic distance between any pair of heavy atoms was less than 0.45 nm. Therefore, the contact metric adopted in this study is based on a heavy-atom–heavy-atom contact definition using the minimum interatomic distance, rather than a residue–residue approximation based on Cα atoms or other single representative atoms.

Hydrogen bonds (HBs) were identified with 'gmx hbond', using the geometric criteria of a donor–acceptor distance $< 0.35$ nm and a donor–hydrogen–acceptor angle $> 150°$. The total potential energy of the system was extracted using 'gmx energy'.

To probe large-scale correlated motions, we performed essential dynamics (ED) analysis, mathematically equivalent to principal component analysis (PCA) [38–40]. The analysis was based on the 3D Cartesian coordinates of all Cα atoms, constructing a covariance matrix of positional fluctuations ($3N \times 3N$, where N is the number of atoms) to capture correlated displacements, which was subsequently diagonalized to obtain eigenvectors representing the principal modes of motion. The first two principal components (PC1 and PC2) were extracted to describe dominant motion patterns, and Boltzmann-weighted free energy landscapes (FELs) were constructed by projecting the trajectories onto the PC space, revealing major conformational barriers and preferences defined by Cα fluctuations. To visualize collective motions along the principal components, eigenvector fields were generated using gmx covar and gmx anaeig and rendered as porcupine plots in PyMOL [41], depicting the displacement vectors of each Cα atom along PC1.

It is important to note that RMSD was computed separately for each of the three independent replicate trajectories to capture local structural variations, whereas RMSF, NNCs, hydrogen bonds, potential energy analysis, ED/PCA, and FEL

construction were performed on the concatenated trajectories of all three replicates, thereby enhancing statistical sampling and providing a more comprehensive view of the global conformational dynamics.

## 2.4. Neural relational inference

To investigate distal residue interactions and potential allosteric communication pathways in SIR2, we employed the Neural Relational Inference (NRI) framework [20,21]. NRI is a graph-based deep learning model that infers latent dynamic interactions between entities via a variational autoencoder (VAE) architecture. The encoder learns a probabilistic latent graph representing residue–residue interactions, while the decoder reconstructs subsequent molecular conformations from the latent graph and initial conditions, enabling the identification of information propagation patterns across spatial and temporal scales in high-dimensional dynamical systems such as protein motions.

In this study, NRI training and inference were performed using three independent 3-μs trajectories from both apo and CNA-bound systems. Each trajectory was initialized with a distinct velocity seed and treated as an independent experimental input. The latent interaction graphs inferred from the three trajectories were subsequently averaged to generate a robust final network, reducing statistical noise and sampling bias inherent to individual trajectories and ensuring reproducibility of the inferred residue communication network.

All trajectories were aligned to their respective initial backbone structures to remove overall translational and rotational motion. Given the large number of residues in full-length SIR2, training NRI on the complete coordinate set would incur high computational costs and convergence difficulties. To address this, we applied a rule-based sparse sampling strategy [42] commonly used in high-dimensional dynamical systems to reduce dimensionality while preserving major conformational modes. Approximately 50% of residues were systematically selected using a "spatially uniform sampling" strategy, which substantially reduced system complexity while preserving the essential dynamical features. To verify the reliability of this approach, we further applied a "reverse sampling" procedure to the remaining 50% of residues and reconstructed the NRI model. The resulting interaction networks and principal communication pathways closely matched those obtained from the initial sampling, demonstrating that this sparse-sampling strategy reliably and faithfully captures the dominant dynamics of the system.

Each trajectory originally contained ~300,000 frames and was uniformly downsampled to 5,000 frames to reduce memory and computational costs. Coordinates and velocity features were normalized to a maximum absolute value of 1. The downsampled trajectories were segmented into overlapping time windows of 50 frames with a stride of 100 frames, allowing the model to capture both short- and long-range temporal dependencies. The NRI model was trained using the Adam optimizer (initial learning rate 0.0005; decayed by a factor of 0.2 every 200 epochs; total of 500 epochs; batch size = 1). The encoder was implemented as a graph neural network (GNN) with variational reparameterization, and the decoder as a multilayer perceptron (MLP) predicting Gaussian-distributed displacements. The objective was to minimize the mean squared error (MSE) between predicted and true trajectory segments. The model was adapted from the open-source implementation by Zhu et al [21].

Upon training, predicted residue–residue interaction graphs were extracted from the latent space and visualized using Cytoscape [43]. The shortest communication paths in the graphs were computed with Dijkstra's algorithm [44] to identify dominant long-range information transfer routes, revealing how CNA binding may reorganize distributed interactions to induce distal conformational changes.

## 2.5. Allosteric pocket prediction and virtual screening

Potential allosteric binding pockets in SIR2 were initially predicted using fpocket [45], which identifies putative ligand-binding regions based on cavity geometry and physicochemical properties. To enhance the functional relevance of pocket selection, the fpocket-predicted cavities were integrated with residue centrality data derived from the NRI network model. Regions showing significant overlap with high-centrality residues were prioritized, ensuring that selected pockets have potential regulatory significance at both structural and network levels.

Compounds were sourced from the ZINC20 [46] database (https://zinc20.docking.org/). From the curated 2D physico-chemical subset (https://files.docking.org/2D/), three fragment-sized libraries (AAAA, AAAB, AAAC) were obtained, totaling 8,852 purchasable compounds with reliable chemical properties. The choice of fragment-sized molecules was deliberate: their size and chemical simplicity are suitable for exploring potentially shallow or narrow allosteric grooves, while maintaining chemical diversity and significantly reducing computational screening costs. Redundant or structurally unstable entries were removed using filter_frags.py, followed by diversity optimization with diverse_picker.py, yielding a reduced set of 147 representative fragment molecules.

All compounds were converted to 3D conformations and protonated/standardized at physiological pH. Molecular docking was performed with AutoDock Vina, using a cubic search space with 40 Å edge length and 1.0 Å grid spacing, centered on the centroid of the predicted pocket. Docked poses were ranked by binding affinity, and the highest-scoring conformations were selected for subsequent structural interpretation and mechanistic analysis.

## 3. Results

### 3.1. Carba-NAD binding induces selective conformational repacking with enhanced thermodynamic stability

To investigate the impact of CNA binding on SIR2 conformational properties, we first compared the overall conformational dynamics in the apo and CNA-bound states. Each system was simulated with three independent 3-μs MD trajectories, and backbone RMSD was subsequently analyzed (Fig 2). All trajectories gradually reached equilibrium during the initial phase and remained stable thereafter, although the relaxation timescales differed between systems. Compared with the apo state, the CNA-bound system exhibited larger RMSD fluctuations and a higher mean RMSD. In some replicates, we also observed discrete RMSD jumps, consistent with rare conformational transitions or metastable state switching within the system.

At first glance, these results might suggest structural instability. However, a more nuanced interpretation is that cofactor binding reshapes the conformational energy landscape—opening new dynamic regions while preserving the overall fold. From this perspective, the increased flexibility reflects a redistribution of accessible conformational states rather than a loss of stability. To further support this view, we analyzed not only global statistics but also time series of native contacts (NNCs), solvent-accessible surface area (SASA), radius of gyration (Rg), and hydrogen bond number (NHB) (Fig A in S1 Text). Based on these data, we systematically quantified key descriptors of structural compactness and energetic stability, including NNC, SASA, Rg, NHB, and total potential energy (Table 1; values in parentheses indicate standard deviations).

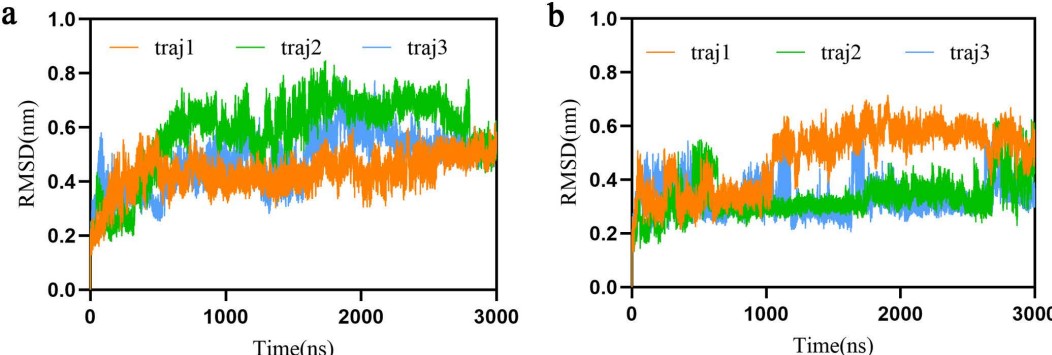

**Fig 2. Backbone RMSD of three independent trajectories for the apo and CNA-bound states.** (a) Apo state; (b) CNA-bound state. Overall, CNA binding increases structural fluctuations and introduces intermittent conformational transitions, while the global fold of SIR2 remains stable. In the CNA-bound trajectories, the third replicate (orange) exhibits higher RMSD fluctuations due to more pronounced local rearrangements in the α1–α2 region, which were not observed in the other two trajectories. Despite the larger amplitude, these fluctuations remain within the stable conformational space and do not compromise the overall structural integrity.

**Table 1. Analysis of intramolecular interactions in SIR2.**

| Protein | NNC[a] | SASA[b](Å2) | Rg[c](Å) | NHB[d] | Energy[e] |
|---|---|---|---|---|---|
| unbound | 84011.1(896.6) | 15537.3(475.0) | 20.6(0.21) | 190.4(8.1) | -627985 |
| CNA-bound | 83631.9(803.5) | 15735.8(498.5) | 20.9(0.36) | 187.3(8.3) | -628597 |

[a]Number of natural contacts: Native contacts were defined with a 0.45-nm heavy-atom cutoffb.

[b]Total solvent accessible surface area.

[c]Radius of gyration.

[d]Number of hydrogen bonds [47] (donor–acceptor distance < 0.35 nm and angle > 150°).

[e]Total potential energy.

Table 1 summarizes intramolecular interactions of SIR2 in the apo and CNA-bound states. Cofactor binding reduces the number of native contacts (NNC) and hydrogen bonds (NHB), indicating a weakening of internal contacts. Correspondingly, SASA increases and Rg slightly expands, reflecting a more open conformational ensemble. Despite these trends, the total potential energy decreases, suggesting that ligand binding thermodynamically stabilizes the complex to some extent, even while locally relaxing structural compactness.

### 3.2. Rigidification of the β1–α2 loop and distal flexibilization define an activation-competent state

Although CNA binding expands the overall conformational space of SIR2, this increase in flexibility is highly non-uniformly distributed. This raises a key question: does this "local rigidification coupled with distal flexibilization" represent a functional adaptation to support activation? Specifically, we sought to determine whether stabilization of local loops is accompanied by loosening of distal regulatory regions, enabling long-range cooperative effects.

To address this, we compared simulation trajectories before and after CNA binding and performed residue-level RMSF analysis. The results revealed a pronounced "bipolarization" pattern (Fig 3a): while fluctuations in many surface and distal regions increased (indicating enhanced plasticity), motion of the conserved β1–α2 loop (residues 35–63) adjacent to the cofactor-binding pocket was strongly suppressed. This local rigidification has important functional implications and is consistent with previous studies showing that this loop plays a critical role in substrate gating, catalytic regulation, and product release.

To link these dynamic changes to functional architecture, we defined five "distal flexibility hotspots" (S1–S5; see green bars in Fig 3a) based on regions showing the most pronounced RMSF increases in the CNA-bound system. When mapped onto the 3D structure (Fig 3b, c), a clear "rigid core–flexible periphery" architecture emerges: upon CNA binding, the β1–α2 loop contracts and stabilizes, whereas the S1–S5 regions exhibit generally enhanced mobility. This spatial bipolarization reflects a selective redistribution of internal entropy—a subtle structural strategy that maintains catalytic precision through local rigidification while leveraging distal flexibility to propagate allosteric signals.

This dual dynamic architecture—rigidification of the catalytic gate coupled with flexibilization of distal relay regions—suggests the presence of a long-range conformational regulatory mechanism. The β1–α2 loop acts as an "anchoring switch," stabilizing the catalytic core while transmitting perturbations, whereas S1–S5 provide distal adaptability for regulatory responses. Together, these dynamic features define an "activation-competent state," in which the catalytic core remains stable while peripheral regions retain adaptive regulatory space. This dynamic organization underpins the allosteric potential of SIR2 and establishes the basis for "distributed residue communication," which we will explore further in the following network analysis.

### 3.3. Conformational dynamics and free energy landscapes reveal the propensity toward an activation-competent state

Although the previous section revealed the spatial pattern of flexibility redistribution—rigidification of the catalytic β1–α2 loop coupled with distal region flexibilization—it remained unclear whether this "bipolarization" represents a passive

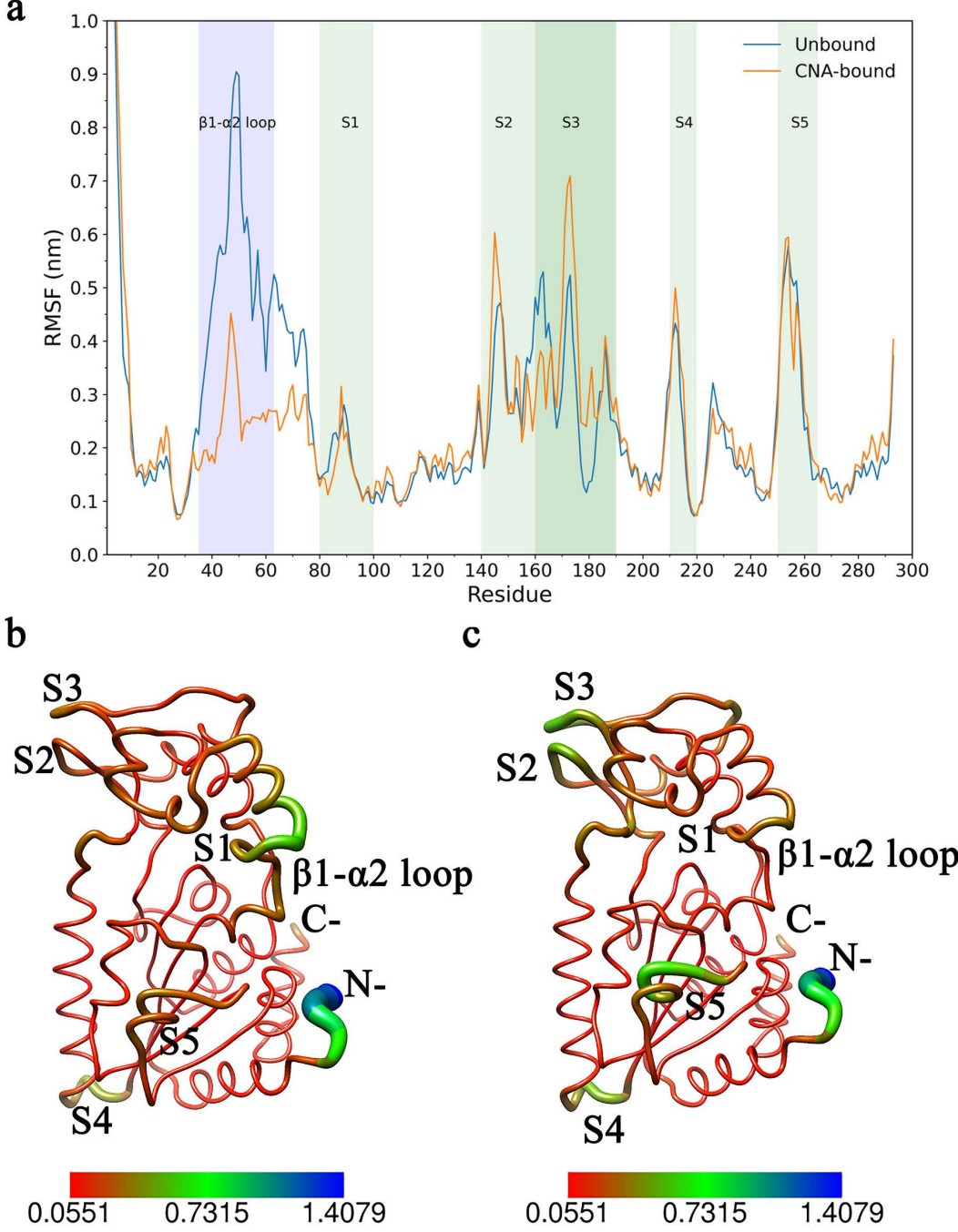

**Fig 3. Selective redistribution of conformational flexibility upon CNA binding.** (a) Residue-level RMSF profiles of SIR2 in the apo state (blue) and CNA-bound state (orange), showing backbone flexibility during MD simulations. The β1–α2 loop (residues 35–63) is highlighted in purple, illustrating pronounced rigidification upon CNA binding. Five distal regions exhibiting increased flexibility in the bound state are defined as S1–S5 and marked with green bars. (b–c) RMSF values mapped onto the 3D backbone structures of apo (b) and CNA-bound (c) SIR2. The color gradient from red (rigid) to blue (flexible) represents relative fluctuation amplitudes, while the thickness of the backbone traces further indicates atomic mobility. Results show that in the bound state, the β1–α2 loop contracts into a rigid conformation, whereas the S1–S5 regions display enhanced mobility, establishing a "rigid core–flexible periphery" architecture.

adaptation to ligand binding or a directed reorganization of the protein's dynamic landscape. To clarify this, we performed principal component analysis (PCA) on the Cα trajectories of the apo and CNA-bound states and reconstructed the corresponding free energy landscapes (FELs). By extracting the slowest and functionally relevant collective motions and weighting them by Boltzmann probabilities, these analyses aim to determine whether cofactor binding enriches substates favorable for catalysis.

The results show that the apo system exhibits nearly isotropic motions with no preferred direction, with the β1–α2 loop displaced outward, corresponding to an open, inactive conformation (Fig 4a). In contrast, the CNA-bound system displays pronounced directional collective motions: the β1–α2 loop contracts inward toward the active site, while distal S1–S5 regions show enhanced outward movements, suggesting that they may function as allosteric signal receivers or regulatory interfaces (Fig 4b).

To reveal the thermodynamic basis of this conformational transition, we constructed FELs along PC1 and PC2, weighted by Boltzmann probabilities (Fig 4c, d). In the apo state, the free energy basin is highly concentrated, with low-energy conformations confined to a narrow region and limited conformational sampling. This compact energy topology indicates low overall flexibility and constrained structural fluctuations in the unbound protein. By contrast, the CNA-bound FEL is markedly broader: low-energy valleys span larger regions along both PC1 and PC2, with increased numbers of dispersed minima, demonstrating that the system can stably sample multiple conformations upon ligand binding. The substantial expansion of the energy landscape indicates that CNA binding enhances overall protein flexibility, enabling exploration of a richer conformational space. Taken together, this shift from a compact, monomorphic FEL to a more open, polymorphic landscape supports the notion that cofactor binding biases SIR2 toward an activation-competent conformational state.

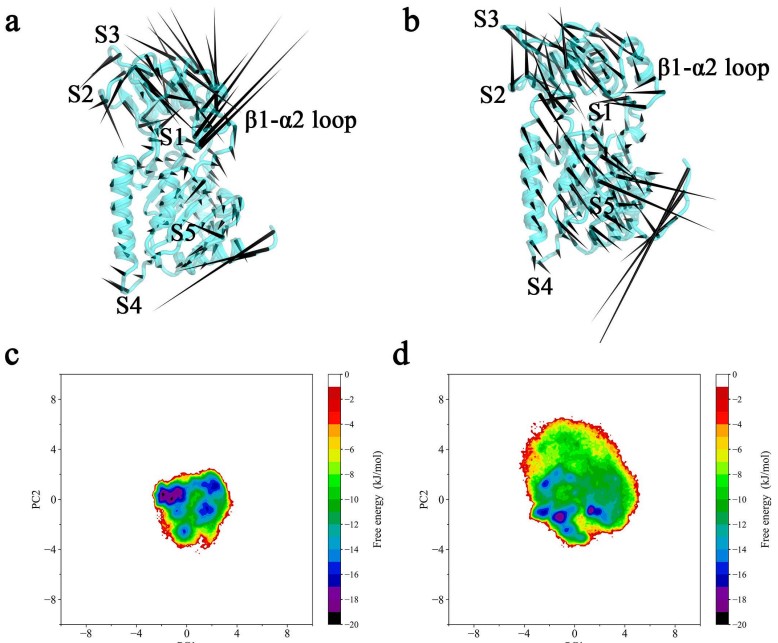

**Fig 4. PCA and free energy landscape analyses reveal CNA-induced conformational tendencies.** (a, b) "Porcupine plots" along the first principal component (PC1) for the apo (a) and CNA-bound (b) systems. Arrows indicate the main motion directions of the β1–α2 loop and distal regions (S1–S5). Results show that cofactor binding shifts the protein from the isotropic "symmetric expansion" mode of the apo state to a polarized "loop-locked, peripheral-relaxed" mode. (c, d) Free energy landscapes projected onto PC1 and PC2 for the apo (c) and CNA-bound (d) systems. Unlike the narrow, centralized energy basin of the apo state, the CNA-bound landscape exhibits a broader and directionally extended basin, reflecting a conformational distribution consistent with the activation-favoring "loop-closure" motion.

### 3.4. Allosteric network remodeling: cofactor binding enhances distal signal propagation

The preceding analyses demonstrated that CNA binding rigidifies the β1–α2 loop while increasing flexibility in other regions of SIR2. A key question, however, remains unresolved: how are these distinct regions coordinated to achieve overall function? Local conformational changes alone cannot explain how catalytic activity is realized, particularly when the catalytic core is stabilized while peripheral regions are simultaneously activated. This observation suggests the existence of a latent intraprotein signaling network capable of converting local energetic stabilization into global information transfer. We sought to determine whether this network is merely perturbed or undergoes a comprehensive reorganization upon cofactor binding.

To address this, we employed the graph-based deep learning model Neural Relational Inference (NRI) to reconstruct residue–residue interaction networks from Cα atom trajectories. Unlike conventional static contact maps, NRI captures how the motion of one residue dynamically influences others, revealing both the directionality and strength of information flow.

The results reveal pronounced network remodeling upon CNA binding. In the apo state, signal coupling is relatively dispersed, long-range connections are weak, and the network is loosely centered on the β1–α2 loop. In contrast, the CNA-bound network displays distinct interaction pathways and locally dense clusters, particularly within distal catalytic regions. This dynamic polarization aligns with the flexibility redistribution described earlier, indicating that conformational changes are accompanied by reorganization of communication pathways. Quantitative analysis of interdomain communication strengths shows that, whereas the apo state relies primarily on the β1–α2 loop as the signaling hub, communication between distal regions—especially S2 and S5—is markedly enhanced upon CNA binding.

To further assess topological changes induced by ligand binding, we compared residue-level network metrics between apo and CNA-bound states (Figs B-C in S1 Text). The CNA-bound network exhibits increases in total edge weight (18.18 vs. 17.30) and average betweenness centrality (0.00505 vs. 0.00440), indicating not only stronger internal communication but also more efficient network organization. Notably, these enhancements are not uniform: several distal residues, including those near positions 212 and 252–254, show pronounced centrality increases, suggesting the emergence of new control hubs beyond the catalytic β1–α2 loop. These nodes, located at hinge or interface regions, likely act as bottlenecks for information flow and provide additional leverage points for allosteric regulation. Overall, these findings indicate that cofactor binding strengthens network cohesion and shifts communication control toward distal hubs.

The network's directed graph representation (Fig 5d, e) visually highlights these structural reorganizations. In the apo state (Fig 5d), the network is dispersed and weakly coupled, with generally thin interdomain edges and no clear dominant pathways. Interactions between the β1–α2 loop and other domains are sparse and weak, indicating a loosely organized internal communication network. By contrast, CNA binding markedly reshapes the network (Fig 5e): the most pronounced enhancements occur among distal regions, where bidirectional edges are substantially thickened, forming a tightly coupled interdomain core. Additionally, interactions between the β1–α2 loop and downstream modules increase relative to the apo state, reflecting its enhanced role in signal transmission upon ligand binding. Collectively, CNA binding transforms a previously diffuse network into a more directional and centralized structure, particularly among downstream domains that form a coordinated dynamical module—consistent with the protein's shift toward an activation-competent conformation.

### 3.5. Cofactor-induced signal propagation connects local anchoring to distal activation

While the previous network analysis revealed that cofactor binding remodels the allosteric communication architecture of SIR2, the precise transmission routes—how regulatory signals propagate from the catalytic β1–α2 loop to distal modules—remained unclear. Although it was known that the catalytic loop becomes rigidified (anchored) while peripheral regions gain flexibility (activated), the physical link between these spatially separated changes was previously undefined. Without specific transmission pathways, the mechanistic connection between local stabilization and distal activation cannot be fully established.

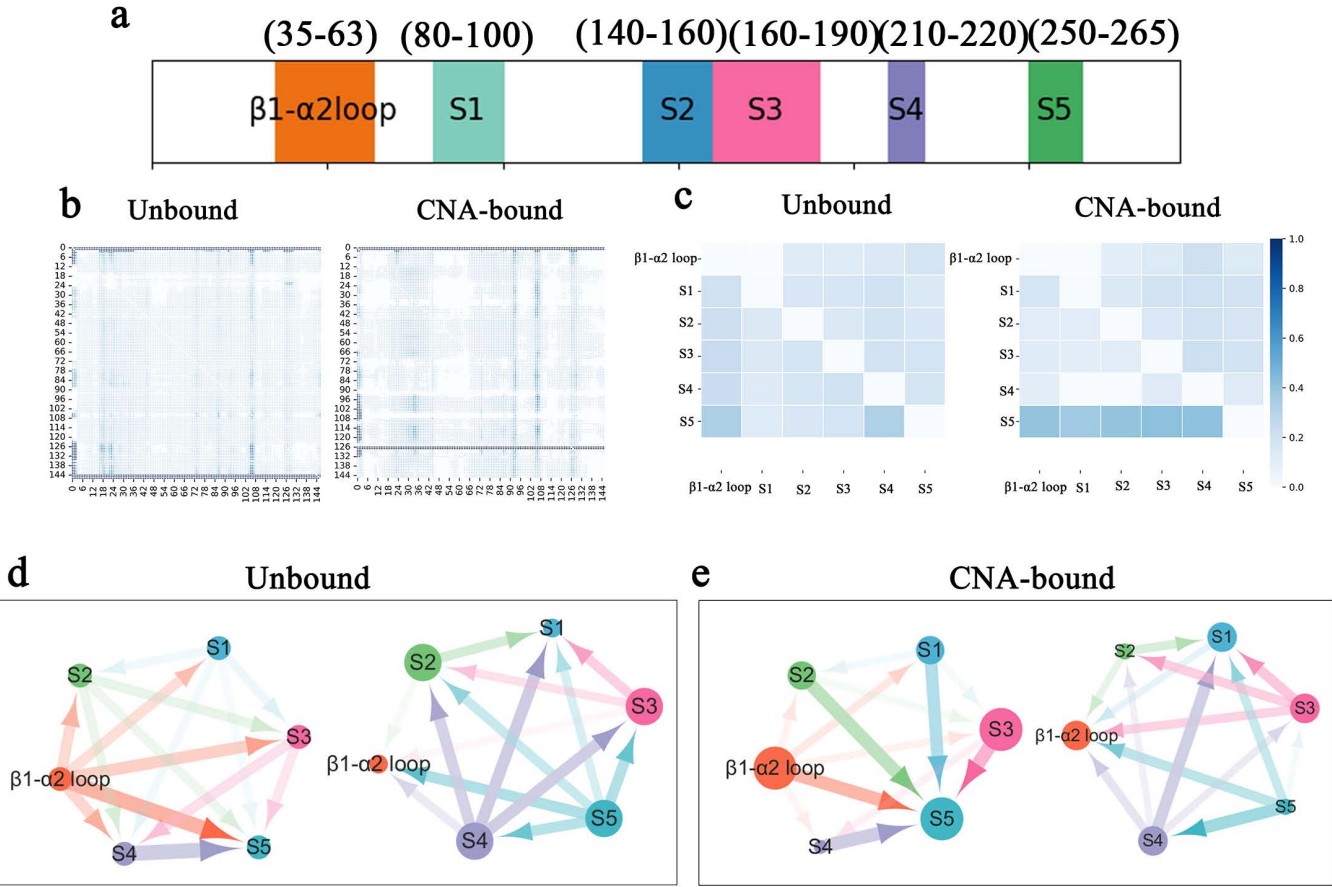

**Fig 5. Cofactor-induced remodeling of the SIR2 residue interaction network.** (a) Schematic domain division used in NRI analysis: catalytic β1–α2 loop (residues 35–63) and five distal regions (S1: 80–100, S2: 140–160, S3: 160–190, S4: 210–220, S5: 250–265). (b) Residue–residue interaction probability matrices inferred by NRI for the apo and CNA-bound states. Color intensity represents the likelihood of inferred interactions. (c) Cross-domain interaction strengths between the catalytic β1–α2 loop and distal modules, highlighting the redistribution of directional influence upon CNA binding. (d, e) Network topology visualized as directed graphs for the apo (d) and CNA-bound (e) states. Node size reflects total interaction strength, and edge thickness indicates directional coupling strength. In the apo state, the β1–α2 loop serves as the primary signaling hub, transmitting strong signals to S1–S5. Upon CNA binding, its centrality decreases, while distal relays S2 and S5 are strengthened, and the overall signaling intensity is slightly redistributed.

To address this, we applied shortest-path analysis to the NRI-derived interaction network, identifying "minimum-resistance" communication routes between the geometric center of the β1–α2 loop and each of the distal loops (S1–S5). These paths represent the most efficient signal-transmission routes within the reconstructed network, allowing quantitative comparison of signal flow before and after cofactor binding.

For visualization, circular layouts were used (Fig 6a and 6b), where each colored path corresponds to one of the five target regions. Nodes are arranged in concentric circles according to their network centrality: highly central nodes occupy the core, while lower-influence nodes reside in the periphery.

Comparison revealed pronounced differences between the apo and CNA-bound states. In the apo state, shortest paths are generally direct and linear, involving only one or two intermediate residues, allowing signals to propagate directly to each target. For example, the path to S2 passes through only two terminal residues (Phe44 and Cys146; Table 2), reflecting a simple, centralized broadcast pattern.

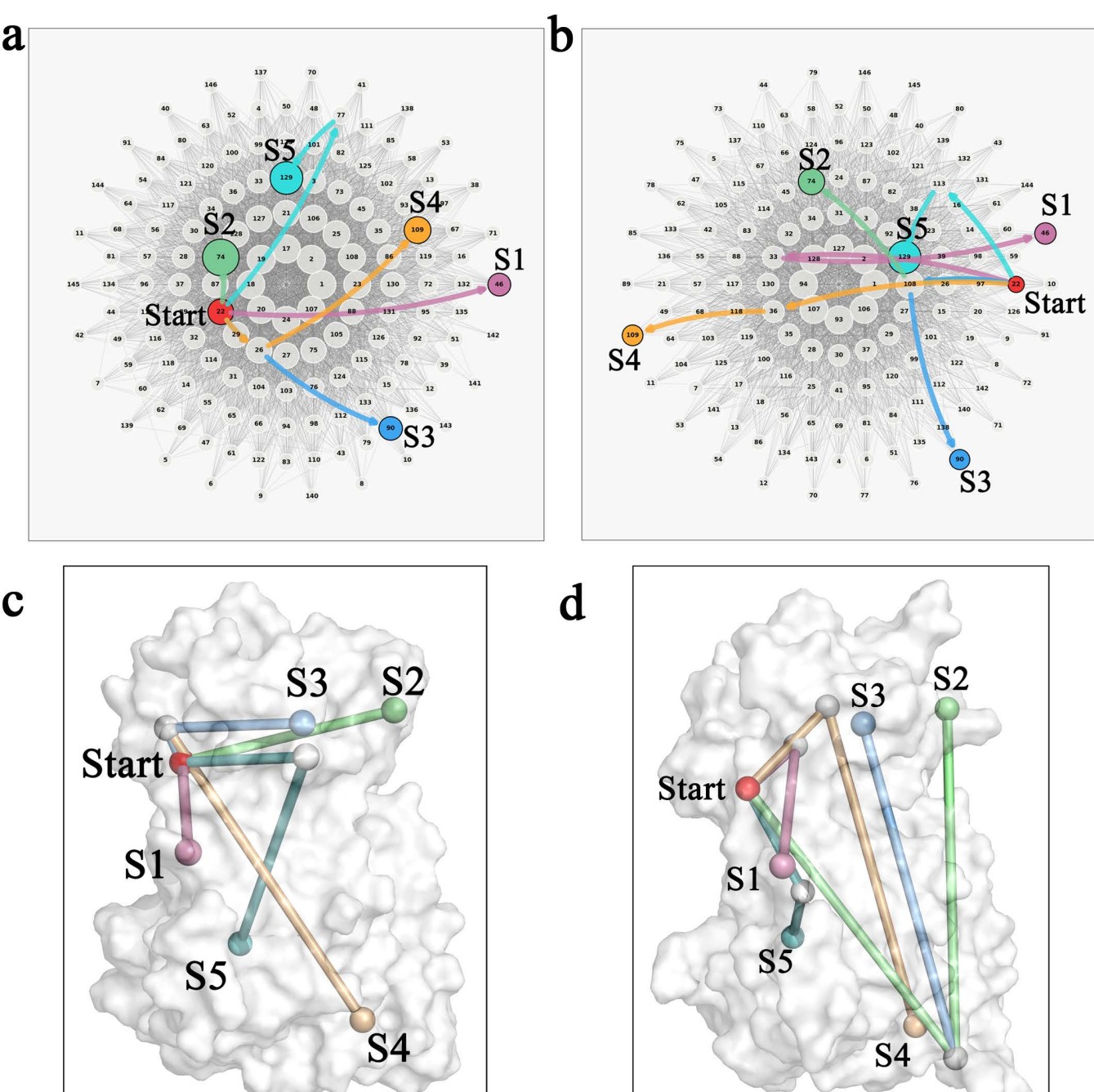

**Fig 6. Topology and spatial distribution of shortest signal-transmission paths in SIR2 before and after cofactor binding.** (a, b) Circular network diagrams for the apo state (a) and CNA-bound state (b), showing the shortest paths connecting the geometric center of the β1–α2 loop (red node) to the five distal flexible regions (S1–S5). Each path is color-coded: pink, green, blue, orange, and cyan correspond to S1–S5, respectively. Nodes are arranged along concentric circles, with proximity to the center indicating higher network centrality; node size also reflects centrality. Edges represent interactions between residues along the paths, showing only residues involved in the shortest communication routes. (c, d) Mapping of the five paths onto the molecular surface of SIR2 in the apo state (c) and CNA-bound state (d). The starting point is the geometric center of the β1–α2 loop (red sphere), and the endpoints represent the geometric centers of the five distal regions (colored spheres). Paths are depicted as smooth curves connecting the key Cα atoms along each shortest route, with colors consistent with panels (a) and (b). A representative frame is shown to illustrate the spatial organization.

**Table 2. Residue-level composition of shortest allosteric communication paths from the β1–α2 loop to distal modules in unbound and CNA-bound SIR2.**

| Pathway Termini | Unbound | | CNA-bound | |
|---|---|---|---|---|
| | Residue numbers | Residue Designations | Residue numbers | Residue Designations |
| S1 | 44-92 | Phe-Arg | 44-66-92 | Phe-Val-Arg |
| S2 | 44-146 | Phe-Cys | 44-214-146 | Phe-Pro-Cys |
| S3 | 44-52-178 | Phe-Tyr-Lys | 44-214-178 | Phe-Pro-Lys |
| S4 | 44-52-216 | Phe-Tyr-Gln | 44-72-216 | Phe-Phe-Gln |
| S5 | 44-152-256 | Phe-Pro-Lys | 44-224-256 | Phe-Thr-Lys |

Upon CNA binding, the network becomes more hierarchical and distributed. Path lengths increase and involve additional intermediate residues. The centrality of the β1–α2 loop decreases; for instance, the path to S2 extends as Phe44→Pro214→Cys146, introducing Pro214 as a new relay node, while the path to S5 now includes Thr224, absent in the apo state. These new nodes are typically located at hinge or interface regions, acting as mechanical checkpoints that fine-tune signal transmission. Their incorporation not only increases path complexity but also introduces a more nuanced regulatory layer, indicating that cofactor binding establishes additional control points to optimize flexibility and synchrony across domains.

The circular layout further illustrates this decentralization: in the CNA-bound state, path nodes are distributed more broadly around the circle, with centrality decreasing sharply toward peripheral nodes, reflecting rapid reassignment of signal-transmission responsibility within the network.

When these reconstructed paths are mapped onto the SIR2 backbone (Fig 6c and 6d), the spatial roles of new relay nodes become evident. In the apo state, signal propagation follows shallow arcs along the protein surface, avoiding structurally constrained residues. In the CNA-bound state, the inclusion of Pro214 and Thr224 forms deeper, structurally anchored pathways, physically linking the catalytic cleft to flexible distal modules. This remodeling converts the β1–α2 loop from a broadcasting hub into a fixed anchor, while distal relay modules such as S2 and S5 assume primary outward transmission roles.

Overall, these results indicate that cofactor binding not only reshapes local geometry and network topology but also establishes new physical signal-transmission routes. The previously dominant signaling β1–α2 loop is anchored, while newly emerged relay nodes distribute communication across the protein scaffold. This redistribution mechanistically connects "local locking" and "peripheral release," providing a basis for the protein to achieve an activation-competent state. Additionally, the emergence of relay nodes and distal hubs suggests that some nodes may overlap with structurally accessible cavities, potentially serving as candidate allosteric ligand-binding sites.

### 3.6. Identification of druggable allosteric pockets in SIR2

Based on the NRI network analysis, Pro214 and Thr224 were identified as new relay residues along the shortest paths, while residues 252–254 were recognized as high-centrality hubs. We next assessed whether these functionally important nodes coincided with geometrically and physicochemically favorable cavities. This step was not merely a geometric pocket search; rather, it aimed to validate whether the communication hotspots identified by network analysis could serve as feasible allosteric regulatory sites.

Fpocket analysis of the SIR2 structure identified 15 potential cavities (Fig I in S1 Text). Cross-referencing these predictions with the network nodes revealed a notable overlap: Pocket 4 (by fpocket ranking) contains both Thr224 (the newly emerged relay residue) and residue 252 (one of the high-betweenness hubs). The dual correspondence of geometric accessibility and network functional importance renders Pocket 4 an attractive candidate site. Its proximity to distal flexible loops further enhances its potential to accommodate regulatory ligands, linking the structural geometry to the dynamic communication logic uncovered by NRI.

To evaluate the ligandability of this pocket, we selected 8,852 chemically diverse fragment-sized molecules from the ZINC20 database, which were then filtered through multiple redundancy removal steps to yield 147 candidate compounds. Docking these compounds into Pocket 4 using AutoDock Vina showed favorable binding, with docking energies ranging from –5.7 to –6.5 kcal/mol, comparable to previously reported allosteric modulators. The top-scoring ligand, ZINC1569826, reached –6.2 kcal/mol and formed four hydrogen bonds with nearby residues (CYS247, ASN248, SER270, and ASP271), stably occupying the pocket (Fig 7). Pocket 4 has a volume of approximately 220 Å³ and moderate polarity, consistent with typical druggable sites.

Analysis of the top ten docked ligands (Table A in S1 Text) further highlighted the pocket's versatility, showing a combination of hydrogen-bonding, hydrophobic, and polar interactions. The overlap between network-identified relay and hub residues and reproducible ligand interactions supports Pocket 4 as a genuine allosteric site with promising druggability. Functionally, these results suggest that allosteric modulators targeting Pocket 4 could reinforce the "loop-locked—peripheral-released" mechanism observed in our simulations. Unlike the structurally constrained, primarily inhibitory orthosteric NAD⁺ site, Pocket 4 offers an opportunity for SIR2 allosteric activation. In the context of aging biology, where NAD⁺ levels naturally decline, allosteric activators mimicking or enhancing NAD⁺ effects could provide a rational route for designing interventions to delay age-related functional decline.

## 4. Discusstion

Although SIR2 family enzymes have long been implicated in aging, metabolic regulation, and stress response [48], their molecular activation mechanisms remain incompletely understood. It is well established that SIR2 catalytic activity depends on NAD⁺ as a co-substrate, and its binding induces conformational rearrangements near the active site [10,15]. However, how this local event propagates throughout the protein to remodel global structure and dynamic organization—particularly how distal flexible modules are coordinated—remains unclear. This reflects a fundamental paradox in allosteric regulation: in the absence of large-scale structural transitions, how can a local binding event reprogram global conformational distributions and long-range communication patterns? Traditional structural biology provides static snapshots before and after ligand binding but lacks the temporal resolution to resolve these dynamic couplings [49]. To address this gap,

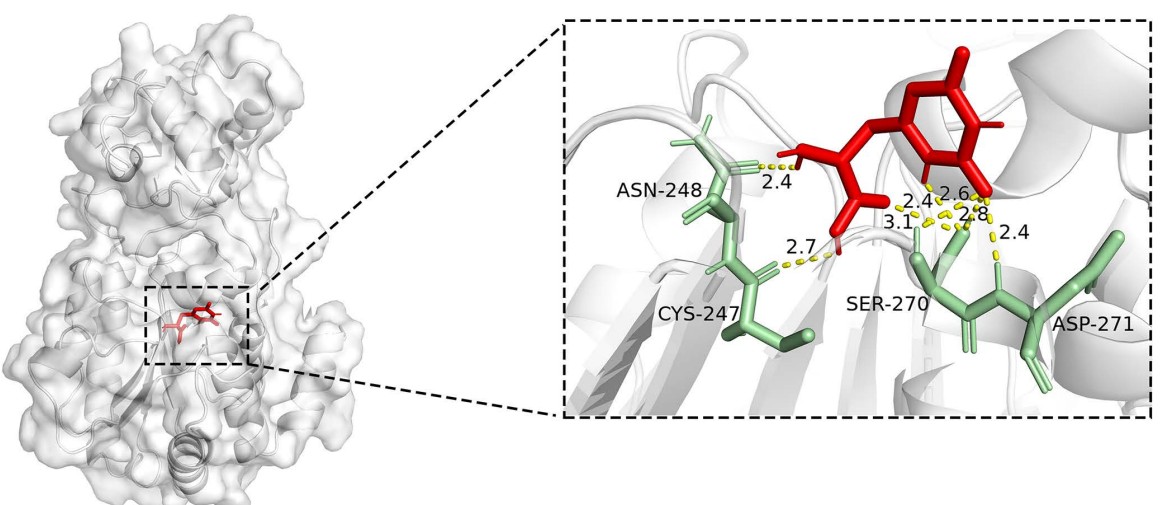

**Fig 7. Illustration of molecular docking results.** The left panel shows the overall binding position of the small molecule (red) within SIR2 Pocket 4 (gray surface representation). The right panel is a close-up of the binding site, highlighting hydrogen-bond interactions (yellow dashed lines) between the ligand and key SIR2 residues (green sticks: ASN248, CYS247, SER270, and ASP271). Numbers indicate hydrogen bond distances in Å.

we integrated molecular dynamics (MD) simulations with graph-based NRI analysis to systematically investigate how CNA binding reshapes SIR2's conformational ensemble and residue interaction network, aiming to decode the mechanism by which cofactors trigger allosteric activation through structural and network reorganization.

Previous studies indicate that conformational redistribution and local rigidification of functional sites are common features of allosteric activation [50]. Guided by this, we first analyzed how CNA binding modulates SIR2 conformational dynamics. Our results show that ligand binding induces a global redistribution of the conformational ensemble, with coordinated changes across multiple structural modules. Most notably, stabilization of the catalytic core coincides with enhanced flexibility of distal loop regions, forming a "core-locked, periphery-released" pattern, suggesting that the protein responds in a distributed manner with coordinated adjustments across the entire backbone. Free energy landscape analysis further supports this view: CNA-bound SIR2 exhibits broader energy basins, indicating increased thermodynamic stability and enhanced conformational accessibility. Consequently, while the active site geometry is optimized, distal regions gain structural plasticity, facilitating transitions into activation-ready states. This surface-level trade-off between local rigidity and distal entropy is a hallmark of allosteric regulation: selective anchoring of functional sites occurs alongside increased peripheral flexibility, optimizing both stability and responsiveness.

Despite these insights into distributed responses, the question remains: how are local and distal changes coordinated? Specifically, does rigidification of the β1–α2 active-site loop couple to distal flexibility through defined communication pathways? To address this, we applied NRI to reconstruct residue interaction networks before and after CNA binding. The results show that ligand binding not only disrupts the single-hub architecture centered on the β1–α2 loop in the apo state but also reprograms global network connectivity and centrality distribution. The β1–α2 loop loses its role as the main broadcast hub, whereas distal segments (S2, S5) become new relay centers. This shift from centralized broadcasting to distributed relays marks a transformation in SIR2's allosteric logic. Biologically, network reprogramming enhances the robustness and spatial coverage of signal propagation while bridging the geometrically stabilized catalytic core with peripheral responsive regions. Thus, CNA stabilizes catalytically favorable local conformations while preparing distal modules for flexible, state-dependent transitions, establishing an activation-ready structural framework.

However, despite network-level reconstruction, the physical pathways of signal propagation across domains remain unclear. Are key relay residues spanning distinct structural domains? Are their positions strategically aligned along allosteric pathways? These questions prompted further analysis of signal transmission routes. To elucidate the physical intramolecular pathways, we performed residue-residue shortest path analysis, identifying minimum-resistance communication chains connecting the catalytic β1–α2 loop to distal modules. Comparison between apo and CNA-bound states shows that CNA expands and diversifies these paths, introducing new relay residues such as Pro214 and Thr224. These residues are strategically located at interfaces or hinge regions, acting as checkpoints to redistribute and finely regulate signal flow. Consequently, CNA converts the β1–α2 loop from a primary signal emitter into a structural anchor, while distal residues take on increased responsibility for outward signal transmission. This reprogramming transforms "local locking and peripheral release" from parallel observations into a causally linked cascade, providing a coherent mechanistic basis for distal activation.

To ensure these findings are not artifacts of the non-hydrolyzable analog CNA, we further simulated NAD$^+$-bound complexes. Across three independent 3 µs trajectories, NAD$^+$ consistently recapitulated CNA's mechanistic features, including β1–α2 loop rigidification, distal flexibility redistribution, and network reorganization. Multi-level analyses of conformational fluctuations, free energy landscapes, and graph networks converge on the same allosteric logic, underscoring the robustness and generality of the proposed activation mechanism independent of the synthetic analog.

Beyond mechanistic insight, our analysis revealed translational potential. The relay residues identified by the network (Pro214, Thr224, and 252–254) overlap with distal cavities detected by fpocket. Fragment docking to this cavity exhibited favorable binding energies and diverse interaction modes, supporting its function as a bona fide allosteric site. Unlike the structurally constrained NAD$^+$ orthosteric site, this distal pocket provides an opportunity for allosteric activation, potentially reinforcing the "loop-locked, periphery-released" mechanism. In aging contexts where NAD$^+$ levels decline, this

pocket offers a rational foundation for designing activators that mimic or enhance cofactor function. The overlap between network-predicted hub residues and geometrically defined cavities (Fig 7), including Thr224 and residues 252–254, forms part of the ligand-binding environment.

In summary, we propose an integrated model of SIR2 allosteric activation: NAD⁺ binding initiates local anchoring at the β1–α2 loop to stabilize catalytic geometry; local rigidity triggers network reprogramming, with distal hubs acting as relays; shortest-path reorganization establishes continuous routes for distal activation; and accessible structural cavities emerge as druggable sites. This hierarchical mechanism—"conformational remodeling, network reorganization, path reprogramming, and appearance of ligandable hotspots"—unifies structural biology with system-level communication logic. From a translational perspective, this model not only elucidates SIR2 activation but also provides a framework for rational design of aging-modulating drugs. More broadly, our study demonstrates how AI-driven network models combined with molecular simulations can decode the latent logic of allosteric proteins, offering a generalizable paradigm for dynamic regulation in enzyme families and allosteric drug discovery in aging and metabolic diseases.

## Supporting information

**S1 Text.** Fig A: Molecular dynamics analyses of SIR2 in the apo and CNA-bound states. Fig B: Total Weight Distribution by Residue. Fig C: Betweenness Centrality Distribution by Residue. Fig D: Conformational stability of SIR2 in the unbound and NAD⁺-bound states. Fig E: Redistribution of conformational flexibility upon NAD⁺ binding. Fig F: Principal component and free energy landscape analyses of NAD⁺-bound SIR2. Fig G: NAD⁺-induced reorganization of the SIR2 residue interaction network. Fig H: NAD⁺-induced remodeling of shortest signal transduction pathways in SIR2. Fig I: Overall distribution of 15 potential binding pockets predicted by Fpocket. Fig J: Control NRI Model Analysis. Table A: Top ten small molecules ranked by molecular docking affinity and their statistical parameters.
(DOCX)

## Acknowledgments

We thank the Biological Big Data and High-Performance Computing Center, School of Agriculture and Life Sciences, Dali University, for providing computational resources and support.

## Author contributions

**Conceptualization:** De-Rui Zhao, Meng-Ting Liu, Li-Quan Yang, Peng Sang.

**Data curation:** Bao-Dan Zhang, De-Rui Zhao.

**Formal analysis:** Bao-Dan Zhang, De-Rui Zhao.

**Investigation:** Bao-Dan Zhang, Peng Sang.

**Methodology:** Bao-Dan Zhang, De-Rui Zhao, Meng-Ting Liu.

**Project administration:** Bao-Dan Zhang, Peng Sang.

**Resources:** De-Rui Zhao, Meng-Ting Liu, Li-Quan Yang, Peng Sang.

**Software:** Bao-Dan Zhang, De-Rui Zhao, Peng Sang.

**Supervision:** Meng-Ting Liu, Li-Quan Yang, Peng Sang.

**Validation:** De-Rui Zhao, Peng Sang.

**Visualization:** Bao-Dan Zhang, Peng Sang.

**Writing – original draft:** Bao-Dan Zhang.

**Writing – review & editing:** Bao-Dan Zhang, Peng Sang.

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
