## [Decision Letter · Decision Letter 0]

1 Aug 2025

Carba-NAD binding activates SIR2 by reshaping conformational plasticity and rewiring long-range allosteric networks

PLOS Computational Biology

Dear Dr. Sang,

Thank you for submitting your manuscript to PLOS Computational Biology. After careful consideration, we feel that it has merit but does not fully meet PLOS Computational Biology's publication criteria as it currently stands. Therefore, we invite you to submit a revised version of the manuscript that addresses the points raised during the review process.

Please submit your revised manuscript within 60 days Oct 01 2025 11:59PM. If you will need more time than this to complete your revisions, please reply to this message or contact the journal office at ploscompbiol@plos.org. Please include the following items when submitting your revised manuscript:

We look forward to receiving your revised manuscript.

Kind regards,

Eduardo Jardón-Valadez

Academic Editor

PLOS Computational Biology

Ilya Ioshikhes

Section Editor

PLOS Computational Biology

**Additional Editor Comments:**

Dear Dr. Peng Sang and Collaborators,

As a complement to our previous letter, we have now received a third review of your manuscript. Once again, we encourage you to carefully consider the reviewers’ comments and suggestions in preparing your revised version.

We look forward to receiving the updated manuscript.

Sincerely,

Eduardo Jardón-Valadez

**Journal Requirements:**

At this stage, the following Authors/Authors require contributions: Baodan Zhang, Derui Zhao, Mengting Liu, Liquan Yang, and Peng Sang. Please ensure that the full contributions of each author are acknowledged in the "Add/Edit/Remove Authors" section of our submission form.

5) Please ensure that the funders and grant numbers match between the Financial Disclosure field and the Funding Information tab in your submission form. Note that the funders must be provided in the same order in both places as well.

**Reviewers' comments:**

Reviewer's Responses to Questions

**Comments to the Authors:**

Reviewer #1: The authors hypothesize that the cofactor binding Carba-NAD, a non-hydrolyzable analog of NAD⁺) leads to long-range allosteric activation of SIR2. They proposed that this activation does not require “large-scale structural changes”, but rather involves redistribution of flexibility, local rigidification, and rewiring of interaction networks within the protein.

To do that they used 1 replica Long-timescale all-atom molecular dynamics (MD) simulations (3 μs for both apo and Carba-NAD bound states), using Neural relational inference (NRI), Principal component analysis (PCA) and free energy landscape (FEL) mapping for collective motions.

Carba-NAD binding “activates” SIR2 by rigidifying the catalytic β1–α2 loop, enhancing flexibility in distal regions, and rewiring residue interaction networks to enable long-range allosteric signaling through newly formed communication pathways.

There are some concerns about the methodology applied.

First, only 1 replica has applied in two systems. Good practices in MD simulations says that at least 3 replicas must be used to perform a standardized analysis of the dynamic of the system. (https://pubs.acs.org/doi/10.1021/acs.jctc.8b00391)

Second, the natural substrate NAD+ has not been simulated neither analyzed, is there any difference between NAD+ and CAN? Does the natural substrate provide different behavior?

Few minor errors must be address:

Reference literature must be standardized, not all the references are with the same style.

Figure 3 cbar has not dimensions.

Trajectories must be provided (https://www.nature.com/articles/s42003-023-04653-0)

Reviewer #2: The manuscript by Zhang et al. describes the computational analysis of SIR2, an NAD+-dependent deacetylase involved in cellular aging. They conducted 3μs Molecular Dynamics (MD) simulations in combination with other techniques and subsequent analyses with the aim of exploring the changes in protein conformation & stability upon co-factor binding. Overall, the research is novel, the experiments conducted appear sound and appropriate, and the figures are clear, informative and well-illustrated with some minor comments.

The authors identify that upon CAN binding, the protein undergoes regional changes in rigidity and motility, with the binding region becoming more rigid and some key distal nodes increasing in flexibility. These observations are indeed novel, but, the manuscript presents some major caveats. Firstly, it would benefit from some greater robustness in the reporting of experimental procedures, as some key methodological details are missing. Secondly, and most importantly, the manuscript would benefit from a more concise writing style. Indeed, the arguments come across as drawn out and overcomplicated, and appear to be repeated throughout the manuscript. Unfortunately, the extent of this phenomenon drastically affects the clarity and ease of reading of the manuscript, thus reducing the impact & accessibility of the research.

This also highlights that there may not be enough data presented to warrant a full article, and whether I) the data should be published under a different format, or II) further experiments (in vitro or in silico) are required before being published as a research article. In particular, the authors state the data presented forms the basis for rational drug design; the inclusion of docking experiments with potential small compounds or proposed peptides for example would greatly increase the impact and pertinence of this research.

Comments:

Lines 16 & 17: There is no need to include these symbols as ‘¶’ only appears once when it indicates authors than contributed equally, while ‘&’ is not used in the author list.

Line 54 & Figure 1 legend: The abbreviation CNA was only defined in the abstract, so should be defined in the introduction. The abstract should be considered independent to the main body of the manuscript with respect to abbreviation defining.

Figure 1: It appears that the unbound and bound images were generated from the same structure, as the only difference between the two panels is the addition of the ligand. If a change is present, it is not visible to the extent suggested in the manuscript to this point, and the figure should be adjusted to better display it.

Line 67: Missing space before “Despite”

Line 107: Which residues are missing? These should explicitly mentioned as the extent and position of these gaps is relevant to the quality of the resulting homology model.

Line 186: Similarly, this residue subset should be detailed as this is a choice which will greatly impact the subsequent analyses. The rationale behind this selection should be explained.

Line 189: What was the downsampling strategy? The chosen parameters could greatly affect the resulting sample sets and analyses.

Table 1: The nature of the number in brackets should be specified. Presumably they are a form of error measurement, but the exact type should be specified.

Figure 5: Panels B & C could benefit from being larger for better inspection of the interaction matrices. More detailed legend for panels D & E should be included, specifically explicitly mentioning the difference between the twinned graphs (i.e. to and from the β1-α2 loop)

Reviewer #3: In this manuscript, the authors propose a novel mechanism by which the binding of a co-factors (Carba-NAD, an analogue for NAD+) induces conformational changes in Sirtuins at 2 levels. It decreases the flexibility around the catalytic site, and at the same time it increases flexibility at distal sites. The authors go further to describe in more detail how the allosteric signal propagates through the structure of the protein. The authors used MD simulations of the apo and cofactor-bound yeast SIR2 to train a Neural Relational Inference (NRI) which is a graph-based deep learning model to infere long range dynamic interactions between protein residues.

My main concern is that the entire manuscript is built based on a single MD simulation, 3 microseconds long for the unbound SIR2 and one more for the co-factor bound SIR2. The authors also do not provide any evidence that the 3 microseconds are enough to converge the motions and this is unlikely to be the case. Therefore it is not clear if the results presented have any statistical significance.

The training of the NRI model is also done based on the same single simulations without any proof or test that the model captures converged long-range interactions in the dynamic network framework.

In addition, there are very small variations between the 2 simulations reported in Table 1 and there are overinterpreted.

**Have the authors made all data and (if applicable) computational code underlying the findings in their manuscript fully available?**

Reviewer #1: **No:** Trajectories must be provided

Reviewer #2: Yes

Reviewer #3: **No:** the MD trajectories are not available

PLOS authors have the option to publish the peer review history of their article (what does this mean? ). If published, this will include your full peer review and any attached files.

**Do you want your identity to be public for this peer review?** For information about this choice, including consent withdrawal, please see our Privacy Policy .

Reviewer #1: **Yes:** Bruno Di Geronimo

Reviewer #2: No

Reviewer #3: No

**Figure resubmission:**

**Reproducibility:**



---

## [Decision Letter · Decision Letter 1]

5 Nov 2025

PCOMPBIOL-D-25-01322R1

Carba-NAD binding activates SIR2 by reshaping conformational plasticity and rewiring long-range allosteric networks

PLOS Computational Biology

Dear Dr. Sang,

Thank you for submitting your manuscript to PLOS Computational Biology. After careful consideration, we feel that it has merit but does not fully meet PLOS Computational Biology's publication criteria as it currently stands. Therefore, we invite you to submit a revised version of the manuscript that addresses the points raised during the review process.

We look forward to receiving your revised manuscript.

Kind regards,

Eduardo Jardón-Valadez

Academic Editor

PLOS Computational Biology

Ilya Ioshikhes

Section Editor

PLOS Computational Biology

**Additional Editor Comments :**

Dear Dr Peng Sang,

We have received the reviewers’ comments and suggestions on your revised manuscript. We encourage you to address their observations and concerns thoroughly. If possible, please highlight the relevance of the neural relational inference methodology in elucidating allosteric regulation, protein dynamics, or protein function based on MD simulation data. We appreciate your efforts to carefully consider and incorporate the reviewers’ feedback in the revised version.

Sincerely,

Eduardo Jardon

**Journal Requirements:**

1) Please amend your detailed Financial Disclosure statement. This is published with the article. It must therefore be completed in full sentences and contain the exact wording you wish to be published.

**Reviewers' comments:**

Reviewer's Responses to Questions

Reviewer #1: The authors have satisfactorily addressed all inquiries raised in the previous round of peer review and have incorporated the suggestions provided by the referees. The authors have addressed all inquiries raised during the first round of peer review and have incorporated all referees’ suggestions in a thorough and constructive manner. It now follows goof MD practices applying 3 replicas. The proposed revisions enhance clarity and strengthen the evidentiary basis of the conclusions, fitting with the PLOS Computational Biology standards. I would like to congratulate again the authors for this complete research.

Reviewer #3: I appreciate very much the effort the authors put into addressing my original comments. Indeed, the revised manuscript contains now a significantly increased amount of data which the authors claim to support their main conclusions. The authors provide now 3 independent MD simulations for each condition and they also include simulation with the naturally occurring co-factor NAD+.

However, it remains unclear how the authors used these new data. With the exception of Figure 2 in which they plot the RMSD individually for each trajectory, it remains unclear how the authors calculated all other parameters using all data they generated. E.g. they do not specify whether they used the ensemble of simulations to calculate the numbers reported or they present averages from the 3 independent replicas. If the latter, error bars with ranges are missing in plots and table. Without a clear explanation of how the authors incorporated all simulations in their analysis, it is impossible to assess the reproducibility and variability between different simulations.

Even if the authors put some effort into improving readability, the paper is still quite bulky and hard to read but in the same time its missing important methodological details.

Some more specific comments:

From the Methods, it remains unclear whether the independent simulations were started after the equilibrations by just re-assigning the velocities or if the entire equilibrations were performed for each replica.

It is not clear what Zn2+ parameters were used

Also in Methods: the authors mention they calculate residue-residue contacts but given the data it seems that what they report are actually atom-atom contacts. Also not clear if only heavy atoms have been used for the calculation.

Was the “FEL” abbreviation defined prior to its use at page 8 in the manuscript?

In the sparse sampling approach for the NRI model, the authors use only half of residues,. This seems rather arbitrary. How do the author justify the selection of 50% of the residues ? They do not present any data to show that 50% would be sufficient to preserve dynamics and no data showing that even sparser choices (e.g. 25%) would not be enough for preserving dynamics anymore.

In Fig 1 it is not clear if the three pictures are having the same orientation. If that is the case, it seems that the colors are not consistent between the 3 pictures. The beta1-alpha2 is shown in purple in the upper illustration and in pink below. Or maybe the labels are not placed in the correct places. In any case, quite unclear.

Font text on graph labels are often too small and hard to read. For example, the RMSD plots in Figure 2 the Y range is too large therefore masking the differences between the replica simulations. Also, from the RMSD plots it is apparent that one of the simulations (in blue) with the co-factor bound behaves differently than the other two. However, the authors do not comment on that

It is not clear what it is shown in Table 1. I would assume that it is the average and standard deviation but this is not specified in the footnotes and values for the standard deviation appear too small. Also not clear how the values were calculated (e.g. considering all simulation replicas or is it an average based on the 3 simulations ?). Could the authors provide the time series of these data as supplementary information ?

Figure 3: how was the RMSF calculated ? Considering all simulations as an ensemble or from the individual simulations ? If the later, the plot from which simulation is shown ?

And the last comment can be applied to principal component analysis and in fact all other analysis (see first comment above)

**Have the authors made all data and (if applicable) computational code underlying the findings in their manuscript fully available?**

Reviewer #1: Yes

Reviewer #3: Yes

PLOS authors have the option to publish the peer review history of their article (what does this mean? ). If published, this will include your full peer review and any attached files.

**Do you want your identity to be public for this peer review?** For information about this choice, including consent withdrawal, please see our Privacy Policy .

Reviewer #1: **Yes:** Bruno Di Geronimo

Reviewer #3: No

**Figure resubmission:**
---

## [Decision Letter · Decision Letter 2]

20 Jan 2026

PCOMPBIOL-D-25-01322R2

Carba-NAD binding activates SIR2 by reshaping conformational plasticity and rewiring long-range allosteric networks

PLOS Computational Biology

Dear Dr. Sang,

Thank you for submitting your manuscript to PLOS Computational Biology. After careful consideration, we feel that it has merit but does not fully meet PLOS Computational Biology's publication criteria as it currently stands. Therefore, we invite you to submit a revised version of the manuscript that addresses the points raised during the review process.

We look forward to receiving your revised manuscript.

Kind regards,

Ilya Ioshikhes

Section Editor

PLOS Computational Biology

Ilya Ioshikhes

Section Editor

PLOS Computational Biology

**Reviewers' comments:**

Reviewer's Responses to Questions

**Comments to the Authors:**

Reviewer #3: I appreciate that the authors have put significant effort in addressing all points I raised. The paper has improved to a point at which it could be published. One concern I still have: why do the authors use 4.5 A as threshold for contacts involving hydrogen atoms ? Since bonds involving hydrogen are rigid during the simulations, I do not see the reasoning for including hydrogens in such calculations. Moreover, residue-residue contacts can be overestimated with such an approach. I would advise redoing the contact analysis by just including heavy atoms and keeping the threshold of 4.5.

**Have the authors made all data and (if applicable) computational code underlying the findings in their manuscript fully available?**

Reviewer #3: Yes

PLOS authors have the option to publish the peer review history of their article (what does this mean? ). If published, this will include your full peer review and any attached files.

**Do you want your identity to be public for this peer review?** For information about this choice, including consent withdrawal, please see our Privacy Policy .

Reviewer #3: **Yes:** Vlad Cojocaru

**Figure resubmission:**
---

## [Editor Report · Decision Letter 3]

2 Feb 2026

Dear Doctor Sang,

We are pleased to inform you that your manuscript 'Carba-NAD binding activates SIR2 by reshaping conformational plasticity and rewiring long-range allosteric networks' has been provisionally accepted for publication in PLOS Computational Biology.

Best regards,

Eduardo Jardón-Valadez

Academic Editor

PLOS Computational Biology

Ilya Ioshikhes

Section Editor

PLOS Computational Biology

Dear Dr. Peng Sang,

I am pleased to recommend your manuscript for publication in PLOS Computational Biology. I appreciate the efforts made to improve the original submission.

Before final acceptance, please consider refining the contact threshold used for hydrogen atoms, or alternatively, provide a clear justification for the rationale underlying the selected cutoff value.

Kind regards,

Eduardo Jardón-Valadez

---

## [Editor Report · Acceptance letter]

PCOMPBIOL-D-25-01322R3

Carba-NAD binding activates SIR2 by reshaping conformational plasticity and rewiring long-range allosteric networks

Dear Dr Sang,

I am pleased to inform you that your manuscript has been formally accepted for publication in PLOS Computational Biology. Your manuscript is now with our production department and you will be notified of the publication date in due course.

With kind regards,

Anita Estes
